# Nanobody repertoire generated against the spike protein of ancestral SARS-CoV-2 remains efficacious against the rapidly evolving virus

Natalia E Ketaren[1†], Fred D Mast[2†], Peter C Fridy[1†], Jean Paul Olivier[2†], Tanmoy Sanyal[3†], Andrej Sali[3], Brian T Chait[4]*, Michael P Rout[1]*, John D Aitchison[2,5,6]*

[1]Laboratory of Cellular and Structural Biology, The Rockefeller University, New York, United States; [2]Center for Global Infectious Disease Research, Seattle Children's Research Institute, Seattle, United States; [3]Department of Bioengineering and Therapeutic Sciences, Department of Pharmaceutical Chemistry, California Institute for Quantitative Biosciences, Byers Hall, University of California, San Francisco, San Francisco, United States; [4]Laboratory of Mass Spectrometry and Gaseous Ion Chemistry, The Rockefeller University, New York, United States; [5]Department of Pediatrics, University of Washington, Seattle, United States; [6]Department of Biochemistry, University of Washington, Seattle, United States

*For correspondence:
chait@rockefeller.edu (BTC);
rout@rockefeller.edu (MPR);
John.Aitchison@seattlechildrens.org (JDA)

[†]These authors contributed equally to this work

**Abstract** To date, all major modes of monoclonal antibody therapy targeting SARS-CoV-2 have lost significant efficacy against the latest circulating variants. As SARS-CoV-2 omicron sublineages account for over 90% of COVID-19 infections, evasion of immune responses generated by vaccination or exposure to previous variants poses a significant challenge. A compelling new therapeutic strategy against SARS-CoV-2 is that of single-domain antibodies, termed nanobodies, which address certain limitations of monoclonal antibodies. Here, we demonstrate that our high-affinity nanobody repertoire, generated against wild-type SARS-CoV-2 spike protein (Mast et al., 2021), remains effective against variants of concern, including omicron BA.4/BA.5; a subset is predicted to counter resistance in emerging XBB and BQ.1.1 sublineages. Furthermore, we reveal the synergistic potential of nanobody cocktails in neutralizing emerging variants. Our study highlights the power of nanobody technology as a versatile therapeutic and diagnostic tool to combat rapidly evolving infectious diseases such as SARS-CoV-2.

## eLife assessment

This study presents **important** insights on the impact of SARS-CoV-2 variants on the binding and neutralization of a small library of nanobodies. The authors should be applauded for their comprehensive in vitro and in silico analyses of nanobody targeting of SARS-CoV-2 variants. The evidence supporting the claims of the authors is now **convincing**. This work will be of great interest to researchers in the fields of antibody/nanobody engineering and SARS-CoV-2 therapeutics.

## Introduction

SARS-CoV-2 has infected >40% of the world's population (***COVID-19 Cumulative Infection Collaborators, 2022***)resulting in a devastating loss of life. As the SARS-CoV-2 pandemic enters its endemic phase (***Meng et al., 2023***; ***Pilz and Ioannidis, 2023***), multiple new variants continue to circulate. Since its initial spread, the rapid adaptation of the virus to selective pressures continues to produce variants of concern (VoC), of which the omicron variants presently account for over 90% of current SARS-CoV-2 infections (https://www.cdc.gov). SARS-CoV-2 displays three structural proteins that are potential targets for therapeutic intervention, but the primary focal point of vaccine development and many therapeutic strategies is the spike surface glycoprotein, which the virus uses to gain cell entry by attaching to the host cell angiotensin-converting enzyme 2 (ACE2) receptor (***Jackson et al., 2020***; ***Krammer, 2020***; ***Letko et al., 2020***; ***Polack et al., 2020***). The spike protein trimer consists of three domains: the receptor-binding domain (RBD) on S1 that binds ACE2, the S1 N-terminal domain (NTD) that has a poorly defined function, and the S2 domain that is involved in virus–host cell membrane fusion (***Walls et al., 2020***; ***Jackson et al., 2022***). Glycosylation is most extensive on the NTD and the S2 domain, whereas the RBD is largely glycan free (***Watanabe et al., 2020***; ***Zhao et al., 2020***). Consequently, it is unsurprising that the most antigenic domain on spike is the RBD, where the majority of neutralizing antibodies have been shown to bind. A comprehensive mapping of the epitopes from 1640 neutralizing monoclonal antibodies (mAbs), all targeting the RBD, revealed 12 epitope groups (***Cao et al., 2022***). These data, combined with previous studies mapping the epitopes of antibodies targeting spike, reveal a total of 19 mAb epitope groups, including seven on the NTD (***Wang et al., 2022d***). Very few anti-S2 antibodies have been shown to be effective therapeutic options (***Wec et al., 2020***), likely due to the shielding effect of S2 glycans (***Grant et al., 2020***).

The mechanism by which omicron variants of SARS-CoV-2 (e.g., BA.1, BA.4, BA.5, and XBB) escape the neutralizing abilities of antibodies generated against spike proteins from preceding variants, whether by vaccination or infection, is largely attributed to the extensive number of mutations accumulated in spike (***Greaney et al., 2021a***; ***Starr et al., 2022***; ***Dadonaite et al., 2023***). Compared to wild-type SARS-CoV-2 spike, omicron BA.1 spike has 37 amino acid residue differences, with almost half located in the RBD domain (***Mannar et al., 2022***). The omicron BA.4/BA.5 variants, which have identical spike proteins (***Tegally et al., 2022***), have additional mutations (including the L452R substitution first seen in the delta variant), that render many previously broadly neutralizing antibodies ineffective (***Cao et al., 2022***; ***Hachmann et al., 2022***; ***Wang et al., 2022b***). Of the mAbs that previously received emergency use authorization (EUA) by the FDA for the treatment of SARS-CoV-2 infection, even cilgavimab and bebtelovimab that were, respectively, moderately and highly efficacious against omicron BA.5, are no longer effective against the current circulating variants XBB, BQ.1.1, and related sublineages (***Takashita et al., 2022***; ***Focosi et al., 2023***; ***Imai et al., 2023***). As a result, no mAb therapy is currently approved by the FDA for treatment of SARS-CoV-2 infection (https://www.fda.gov).

Nanobodies, single-domain antibodies derived from a unique heavy chain-only class of llama antibodies, present numerous therapeutic benefits compared to mAbs. Their smaller size and increased stability make them more resistant to denaturation, simpler to produce, and easier to modify in order to adjust properties such as immunogenicity and half-life (***Muyldermans, 2013***). One potential advantage for neutralizing the spike protein is their compact size and distinctive binding attributes, which allow them to access and bind to epitopes that mAbs cannot reach. Consequently, while the antigenic evolution of the spike protein in response to antibodies has largely rendered mAbs ineffective in a therapeutic context, it remains uncertain how this applies to nanobodies. Moreover, the diminutive size of nanobodies enables them to bind concurrently to a single antigen through non-overlapping epitopes, making them well suited for creating nanobody mixtures with the potential for highly synergistic effects (***Fridy et al., 2014***; ***Mast et al., 2021***).

Here, we demonstrate that a subset of our previously published repertoire of nanobodies, generated against spike from the ancestral SARS-CoV-2 virus (***Mast et al., 2021***), retains binding and in vitro neutralization efficacy against circulating VoC, including omicron BA.4/BA.5. We show the power of nanobodies when working synergistically to create potent neutralizing mixtures against the different VoCs. We also predict that a subset of these nanobodies will remain efficacious against the circulating XBB and BQ.1.1 sublineages. Our study underscores the importance and versatility of large, diverse

repertoires of nanobodies, in their potential to create long-term therapeutic options against rapidly evolving infectious agents such as the SARS-CoV-2 virus.

## Results and discussion

### Nanobodies generated against wild-type SARS-CoV-2 spike remains efficacious against delta, and omicron lineages BA.1, BA.4/BA.5, XBB, and BQ.1.1

From the original nanobody repertoire that we generated against SARS-CoV-2 wild-type spike protein (*Mast et al., 2021*), representative nanobodies from all 10 structurally mapped epitope groups that we previously identified (*Mast et al., 2021*; *Cross et al., 2023*), were selected for SARS-CoV-2 pseudo-virus (PSV) neutralization assays against the SARS-CoV-2 delta and omicron BA.1 strains (*Figure 1*; *Figure 2—source data 1*). Of the 41 nanobodies tested, 35 remained efficacious against at least one variant, where 28 neutralized delta, 23 neutralized omicron BA.1, and 15 neutralized both. The RBD groups I, I/II, II, I/IV, and IV – where nanobodies whose epitopes could not be distinguished between two groups are demarcated I/II and I/IV – and the anti-S2 groups (groups IX and X) contain a high number of nanobodies that neutralized delta. We note that $IC_{50}$s are not directly comparable across different experimental setups because measured values are highly dependent on the experimental conditions. For this reason, we included other published nanobodies as benchmarks in our original publication (*Mast et al., 2021*) and have subsequently maintained standard experimental conditions. Additionally, for computational epitope modeling, we selected nanobody candidates using a series of experimentally obtained structural restraints, as described in *Mast et al., 2021*.

These observations align with data from mAbs approved by the FDA, where nanobody epitopes from groups I, II, and IV (*Figure 1B*) overlap with the epitopes of five mAbs effective against delta, categorized into two of three classes on RBD: class 1 (etesivimab, casirivimab, and amubarvimab; *Greaney et al., 2021b*; *Li et al., 2021*; *Planas et al., 2021*; *Takashita et al., 2022*; *Cox et al., 2023*) and class 3 (imdemivab and bebtelovimab; *Cox et al., 2023*; *Figure 2*). Notably, most of our group I and II nanobodies neutralized delta better than wild-type (*Figure 1*), paralleling observations made with the mAb etesivimab (*Wang et al., 2022c*, *Cox et al., 2023*). In comparison, groups I, I/II, I/IV, V, VII, VIII, and the anti-S2 nanobodies contained the majority of omicron BA.1 neutralizers, though here the neutralization potency of many nanobodies was generally decreased tenfold compared to wild-type. This decrease in neutralization potency largely correlates with the accumulation of omicron BA.1-specific mutations throughout the RBD, which potentially alter the nanobodies' binding sites, weakening their interaction with the BA.1 spike (*Figure 1B*). Concomitantly, groups I, I/II, I/IV, and the anti-S2 groups contain nanobodies able to neutralize both delta and omicron BA.1. These results demonstrate the effectiveness of our original nanobody cohort against delta and omicron BA.1, targeting all major regions of spike.

Of the nanobodies that neutralized both delta and omicron BA.1, representatives from each of the nanobody epitope groups were selected for surface plasmon resonance (SPR) analysis, where S1 binders with mapped epitopes that neutralized one or both variants well were prioritized. SPR-binding assessments to the spike S1 domain or RBD of delta revealed a pattern: nanobodies maintaining binding affinity generally also neutralized the virus with a statistically significant correlation between binding affinity and neutralization efficacy (Pearson's correlation coefficient: 0.71, p-value: 0.01; Spearman's rho: 0.63, p-value: 0.07). However, this correlation was not statistically significant for omicron BA.1 (Pearson's correlation coefficient: 0.27, p-value: 0.31) (*Figure 3A*). Notably, while some nanobodies bound to the variants, they did not consistently neutralize them, suggesting additional factors influence neutralization beyond mere binding. We also tested this cohort for binding to omicron BA.4/BA.5 RBD using SPR, revealing that almost all the nanobodies in group I, I/II, II, and I/IV retained binding to omicron BA.4/BA.5. Based on our previously mapped nanobody epitopes on spike (*Mast et al., 2021*), these four groups appear to at least partially overlap with the spike epitope of the potent omicron BA.4/5 neutralizing mAb formerly FDA-approved, bebtelovimab (*Figure 2*; *Focosi et al., 2023*). The nanobodies that retained binding to omicron BA.4/5 were further tested for binding against the omicron XBB and BQ.1.1 lineages in addition to S1-1 and S1-23, revealing nanobodies that bound omicron BA.4/BA.5, aside from S1-39, also bound omicron variants XBB and BQ.1.1 (*Figure 3A*).

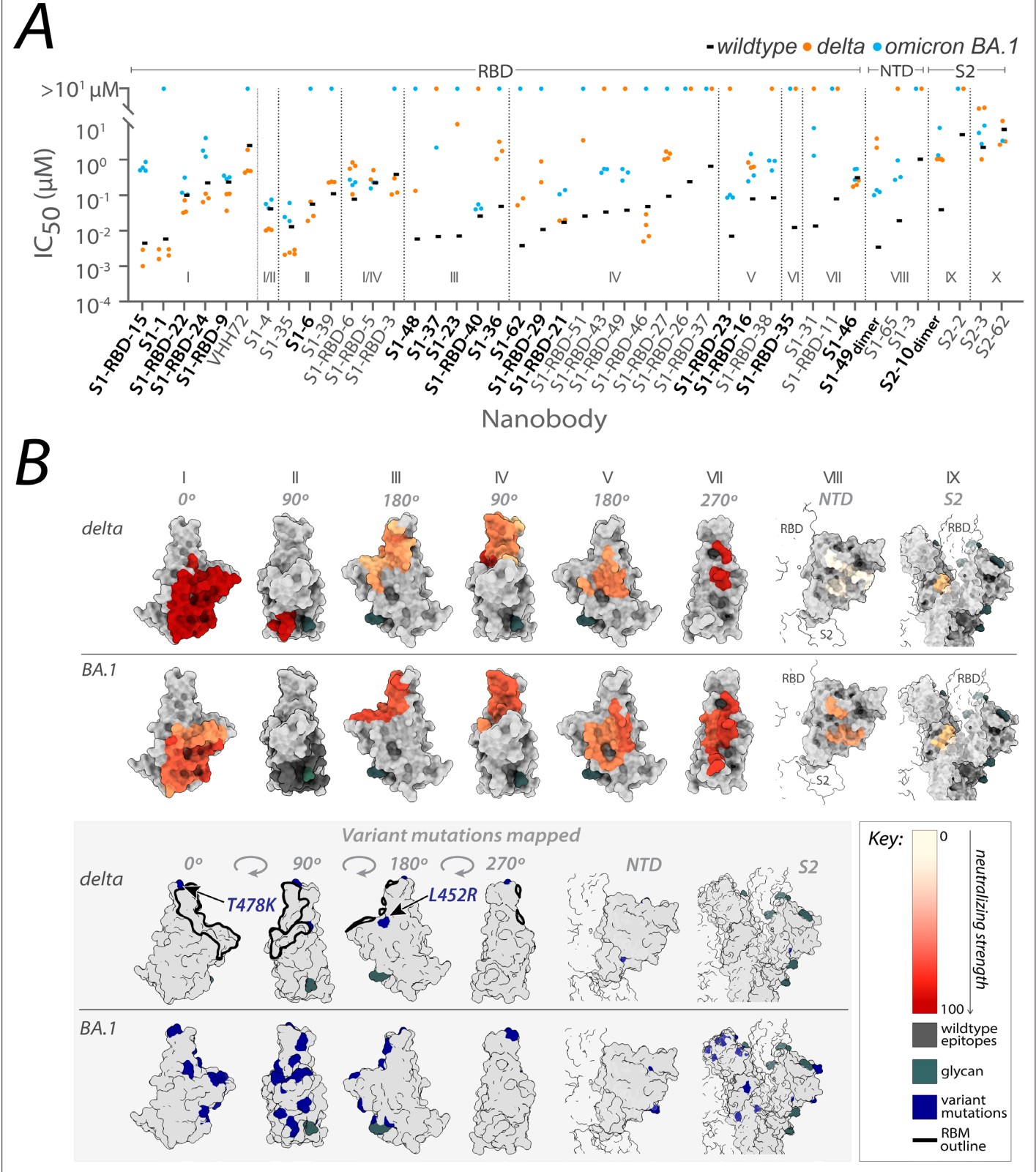

**Figure 1.** Nanobody repertoires generated against wild-type SARS-CoV-2 remain efficacious. Nanobodies targeting the S1-RBD, S1 non-RBD, and S2 regions of spike effectively neutralize lentivirus pseudotyped with delta and omicron BA.1 SARS-CoV-2 spikes (PSV) from infecting angiotensin-converting enzyme 2 (ACE2)-expressing HEK293T cells. (**A**) The half-maximal inhibitory concentration ($IC_{50}$) is reported for the indicated nanobodies against wild-type (***Mast et al., 2021***), delta, and omicron BA.1 PSV. These values are summarized in ***Figure 2—source data 1***. Nanobodies are

*Figure 1 continued on next page*

*Figure 1 continued*

grouped by epitope and arranged within each epitope by neutralization efficacy against the wild-type PSV. $n \geq 4$ (**B**) The structural differences in the receptor-binding domain (RBD) of the delta (PDB ID: 7SBO) and omicron BA.1 (PDB ID: 7T9K) variants are depicted. Nanobody epitopes are heat-mapped ranging from pale white (epitopes with weak neutralization against SARS-CoV-2) to dark red (indicating strong neutralization). Boxed in gray are mutations specific to each variant mapped in blue on the aforementioned structures. The nanobodies that contributed to epitope mapping are in bold in panel A. The color bar scale for each epitope is the neutralizing strength of each nanobody epitope, calculated as the normalized −log10 ratio of nanobody binding (IC$_{50}$) to variant versus wild-type SARS-CoV-2 Spike S1. For groups with multiple nanobodies, the average −log10 (IC$_{50}$) is first calculated for the nanobodies within that group, then normalized to a neutralization score within the 0–100 range using the min and max average −log10 (IC$_{50}$) for that group. A higher score indicates more potent neutralization of the variant relative to the wild-type. All structural representations were created on ChimeraX (*Pettersen et al., 2021*).

The remaining six nanobody groups tested, including the major nanobody groups III and IV, showed no detectable binding to omicron BA.4/BA.5/XBB/BQ.1.1 RBD, save for RBD-40 (group III) and RBD-46 (group IV), which exhibited a ~40- and ~20-fold decrease in affinity, respectively, compared to wild-type. The diminished binding likely results from the concentration of omicron BA.4/BA.5/XBB/BQ.1.1-specific mutations that overlap with the epitope regions of the affected nanobodies. These mutations sufficiently alter the epitopes, to either abolish or significantly reduce binding (*Figure 3B*). Interestingly, one nanobody, S1-46 (group VII, *Figures 1B and 3B*) retained wild-type binding affinity to RBDs from delta and all four omicron variants tested (*Figure 2*). S1-46 binds a region on spike that is conserved across all variants to date, but which may be relatively inaccessible unless the RBDs are in the 'up' conformation. The epitope of S1-46 is not targeted by any of the mAbs that previously received EUA by the FDA (*Cox et al., 2023*). Guided by the SPR results against omicron BA.4/BA.5 RBD, nanobody neutralization of live BA.5 was performed using the plaque reduction neutralization test as previously described (*Mast et al., 2021*). All five nanobodies tested neutralized the live omicron BA.5 live virus (*Figure 4*), corroborating our SPR observations. This indicates that our nanobody repertoire generated against wild-type spike has retained efficacy against omicron BA.4/BA.5.

## Impact of spike structural differences across variants on nanobody binding and neutralization potency

The numerous structural differences observed in the spike protein of the delta and omicron sublineages compared to wild-type, have enabled these variants to escape the neutralizing effect of most mAbs, with only one clinically approved mAb retaining potency against omicron BA.4/BA.5 (Takashita et al., Cox et al., Focosi et al.). These structural differences have also affected our nanobody cohort, likely playing a role in the differential binding and neutralizing abilities observed against the three tested variants (delta, omicron BA.1, and omicron BA.4/5). Notably, all but one of our nanobodies in groups I, I/II, and II, displayed *better* neutralizing ability against delta compared to wild-type (*Figure 1*). It is possible that this difference is due to the delta spike trimer's preference for the up orientation of its RBDs, resulting from the increased dynamics in its S1 domain compared to wild-type (*Wang et al., 2022a*); this more 'open' state likely allows greater accessibility to the epitopes of these nanobody groups, for net stronger binding. Additionally, the absence of delta-specific mutations within the epitope regions of groups I and II preserves their integrity for nanobody binding (*Figures 1B and 3B*). For the remaining RBD nanobody groups, we are likely observing the impact of the two delta mutations T478K and L452R on nanobody binding and neutralization. These two mutations lie within the mapped epitope regions of group III, IV, and V nanobodies (*Figures 1B and 3B*), which may alter the epitopes of many of these nanobodies enough to negatively impact both binding and neutralization. Importantly, the L452R mutation seems to be a key substitution that contributes to reduced or abolished neutralizing abilities of many mAbs (*Laurini et al., 2021*; *Starr et al., 2021a*, *Starr et al., 2021b*). Antibodies that rely on L452 to create hydrophobic interactions within their epitope will most likely have their binding greatly disrupted with a substituted arginine. Coupled with the stronger affinity between ACE2 and spike caused by L452R (*Motozono et al., 2021*; *Yan et al., 2022*), this substitution can greatly lessen the neutralization ability of nanobodies and antibodies targeting this region.

Unlike wild-type, the spike trimer of omicron BA.1 favors a one-RBD up confirmation (*Zhao et al., 2022*). Though this conformation may facilitate access of our nanobodies to their epitopes on RBD, unlike delta, omicron BA.1 contains many unique mutations distributed throughout the RBD domain

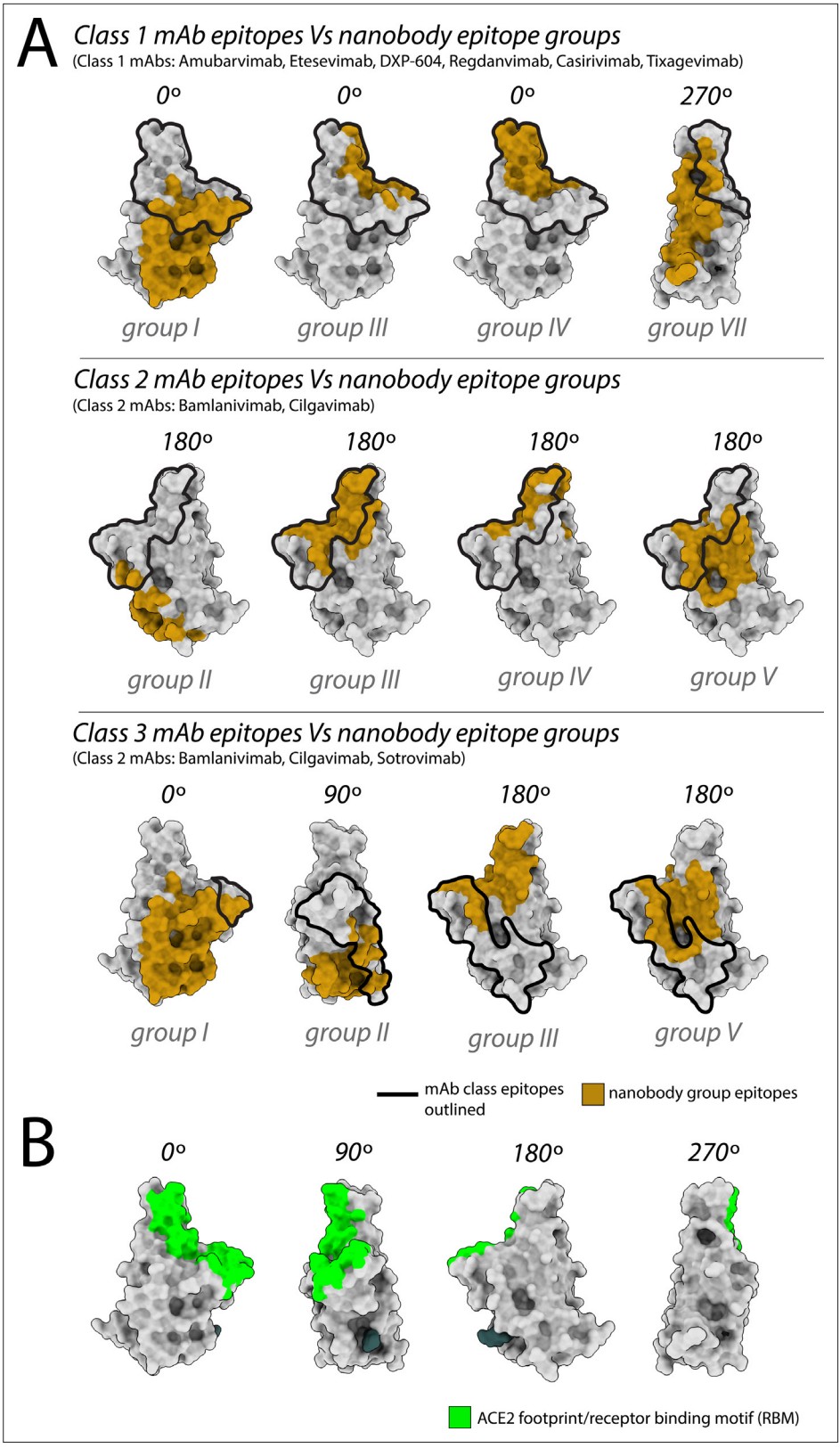

**Figure 2.** Nanobody epitope groups and mAb epitope classes mapped on receptor-binding domain (RBD). (**A**) Nanobody epitope groups overlapping with the three mAb epitope classes (classes 1, 2, and 3). Nanobody groups are highlighted in gold, while mAb class footprints are outlined in black. mAb epitopes are taken from Cox et al. (**B**) A single RBD subunit with the angiotensin-converting enzyme 2 (ACE2) footprint/RBM mapped in green.

*Figure 2 continued on next page*

*Figure 2 continued*

All epitopes are represented on the structure of wild-type RBD (PDB ID: 6M0J). All structure representations were generated using ChimeraX (Pettersen et al.).

The online version of this article includes the following source data for figure 2:

**Source data 1.** Nanobody binding and neutralization characterization; related to *Figures 2 and 3*.

that overlap or flank the epitopes of almost all our nanobody groups (*Figures 1B and 3B*). These mutations likely alter the epitopes of our nanobodies and may be the major contributing factor to the observed decrease in binding affinity and neutralizing potency of our nanobodies against omicron BA.1 (*Figures 1A and 3A*). However, the following cohort of nanobodies retained binding and neutralizing ability similar to wild-type against omicron BA.1: S1-RBD-22, S1-RBD-9 (both group I, class 1), S1-4 (group I/II, class 3), S1-RBD-5 (group I/IV, class 3), S1-46 (group VII), and members of group X. The additional mutations on the spike of omicron BA.4/BA.5, many of which overlap with our nanobody epitopes, are predicted to further impact the binding of our nanobody repertoire to this variant. For example, omicron BA.4/BA.5 re-introduces the key L452R substitution from delta, which when combined with the accumulated mutations on spike from omicron precursor sublineages, could be responsible for the observed loss of binding of numerous nanobodies in groups III, IV, and V. Furthermore, the differential binding and neutralizing abilities of group I and II nanobodies against omicron variants BA.4/BA.5/XBB/BQ.1.1 may be because of the T376A, D405N, and R408S substitutions, which lie within and near the epitope regions of groups I and II, respectively (*Figure 3B*). Importantly, at least five nanobodies from groups I, I/II, II, and V retained neutralization activity against omicron BA.5 (*Figure 4*), demonstrating the broad specificity of this set of nanobodies.

## Nanobodies that lose neutralization ability can still bind spike

Our SPR experiments largely correlated with the neutralization data: nanobodies that showed binding to spike also neutralized the virus, and where the binding affinity decreased significantly, a loss of neutralization ability was observed. This was seen with the group III nanobodies S1-23 and S1-37 that both demonstrated significantly reduced binding affinity and neutralization potency against both delta and omicron BA.1. This reduction in efficacy is likely attributable to the L452R mutation against delta, and the extensive amino acid changes in omicron BA.1 compared to wild-type (see above). However, we observed instances where nanobodies retained binding to spike yet no longer neutralized the virus. Nanobodies S1-36, S1-39, and S1-RBD-29 showed binding to omicron BA.1 (*Figure 3A*), yet none neutralized the variant in the pseudovirus assay (*Figure 1A*). These nanobodies are in groups III, II, and IV, respectively (*Figure 3B*) – epitope regions that contain numerous omicron BA. 1 mutations. For all three nanobodies, the mutations peppered throughout their epitope space have likely altered the binding landscape, resulting in decreased affinity (~100-fold for S1-36 and S1-39) and ineffective neutralization in vitro. Additionally, S1-RBD-29 binds an epitope that overlaps with the ACE2-binding site, and likely neutralizes the wild-type strain by blocking the ACE2 interaction (*Figure 1B*). Alterations in the omicron BA.1 epitope may have changed the orientation of the nanobody as it binds, negating effective blocking of ACE2 binding and thus neutralization. In contrast, S1-37 binds both delta and omicron BA.1 with similar reduced affinities (>100-fold decrease compared to wild-type), yet only neutralized omicron BA.1 (*Figure 1A*). As mentioned above, the epitope of S1-37 overlaps with the delta L452R mutation (*Figure 1B*), which has impacted the effectiveness of numerous neutralizing antibodies (*Bian et al., 2021*). It is possible that this mutation alone drastically weakens S1-37 binding to delta and consequently virus neutralization. Lastly, S1-RBD-43, whose epitope as a group IV binder is predicted to overlap with the ACE2 binding footprint (*Figures 1B and 3B*), showed binding to delta with equal affinity to wild-type (*Figure 3A*), yet does not neutralize delta (*Figure 1A*). The delta mutation T478K is present within the epitope region of group IV (*Figures 1B and 3B*), and was shown to significantly increase the interaction of delta spike for ACE2 by creating a new salt bridge at the RBD/ACE2 interface (*Cheng et al., 2022*). It may be that, as suggested for S1-RBD-29, the binding orientation of S1-RBD-43 has been altered allowing the RBD to maintain an interaction with ACE2 despite the presence of S1-RBD-43, thus rendering the nanobody ineffective in neutralizing delta.

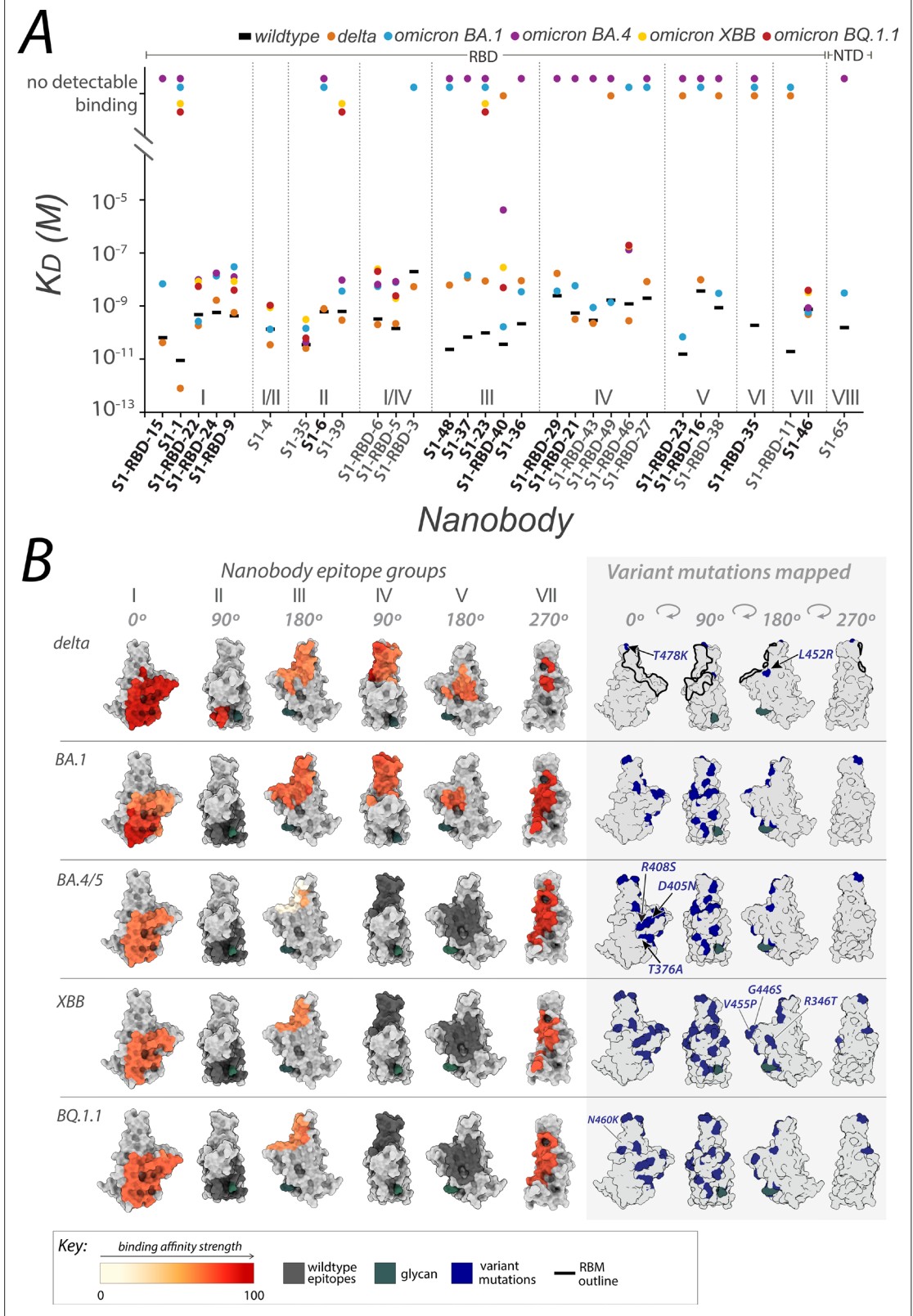

**Figure 3.** Affinities of the nanobody repertoire against SARS-CoV-2 variants. (**A**) Each nanobody is plotted against their affinity ($K_D$) for SARS-CoV-2 Spike S1 from wild-type, delta and omicron BA.1, BA.4, XBB, and BQ.1.1 strains. Kinetic values are summarized in . Nanobodies are characterized into their respective epitope groups as described previously (**Mast et al., 2021**). (**B**) Displayed are the structures of the receptor-binding domain (RBD) of spike delta (PDB ID: 7SBO), omicron BA.1 (PDB ID: 7T9K), omicron BA.4 RBD modeled using AlphaFold (**Jumper et al., 2021**), omicron XBB (PDB

*Figure 3 continued on next page*

*Figure 3 continued*

ID: 8IOU), and omicron BQ.1.1 (PDB ID: 8FXC). The structures feature heat-mapped epitopes of binding, ranging from pale white (weak binding to SARS-CoV-2) to dark red (strong binding to SARS-CoV-2). In the gray box, mutations specific to each variant are highlighted in blue. The nanobodies that contributed to epitope mapping are in bold in panel A. The color bar scale indicates each epitope's binding affinity strength, represented as the normalized −log10 ratio of nanobody binding ($K_D$) of variant versus wild-type SARS-CoV-2 Spike S1. For groups with multiple nanobodies, the average −log10 ($K_D$) for the nanobodies within that group was calculated, then normalized to an affinity score ranging from 0 to 100 using the min and max average −log10 ($K_D$) for that group. Higher −log10 ratios indicate stronger binding of the nanobody to the variant versus wild-type. S1-RBD-16 bound omicron BA.1 and BA.4/5 in ELISA. S1-RBD-11 was not tested against omicron BA.4. S1-65 was not tested against BA.1. Only S1-1, S1-RBD-22, S1-RBD-9, S1-4, S1-35, S1-39, S1-RBD-6, S1-RBD-5, S1-23, S1-RBD-40, S1-RBD-46, and S1-46 were tested against omicron XBB and BQ.1.1. All structure representations were generated using ChimeraX (*Pettersen et al., 2021*).

## Identification of variant-specific epitopes and broadly neutralizing epitope groups

The results of our neutralization assays and affinity measurements revealed that ~1/3 of our original repertoire of 116 nanobodies (*Mast et al., 2021*) generated against wild-type SARS-CoV-2 remain effective binders/neutralizers of the variants tested. Specifically, nanobodies from 11 of the 18 nanobody groups (inclusive of the 10 mapped epitopes) demonstrated efficacy against one or more of the delta, omicron BA.1, omicron BA.4/BA.5, omicron XBB, and omicron BQ.1.1 lineages. The varied efficacy of nanobodies within each group, along with structural modeling, enabled us to expand and further refine our original six structurally modeled RBD epitope groups to a total of 12 (*Figure 5A*). These 12 groups were then contrasted with the three mAb classes containing mAbs previously approved for EUA by the FDA (*Figure 2*). This comparison revealed the following overlaps between our 12 RBD nanobody epitope groups and the three mAb classes: groups I, III, IV, and VII overlap with

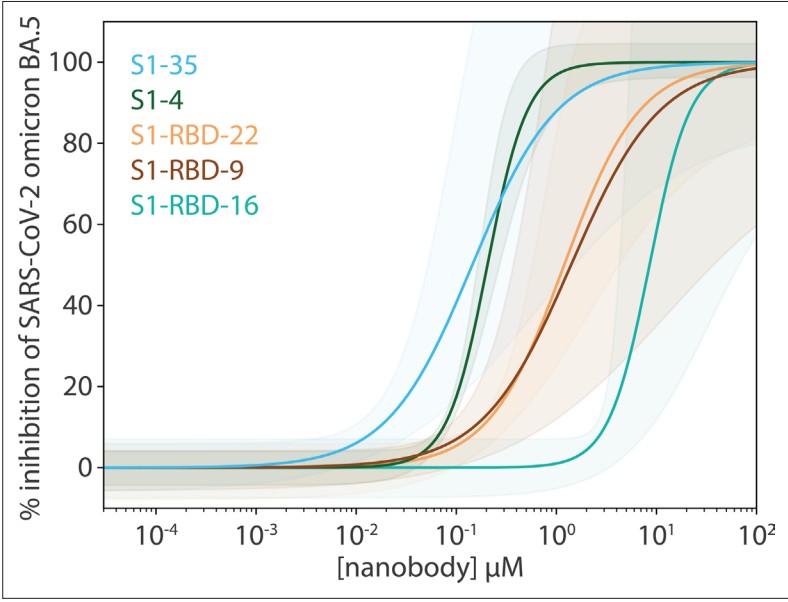

**Figure 4.** Potent neutralization by broadly active nanobodies. Nanobodies targeting the S1-RBD of spike and raised against the original wild-type sequence remain highly efficacious in neutralizing an evolved variant of SARS-CoV-2, omicron variant BA.5. The derived neutralization curves are plotted from the results of a plaque-forming reduction neutralization test with the indicated nanobodies. Serial dilutions of each nanobody were incubated with ~200 SARS-CoV-2 virions for 60 min and then overlaid on a monolayer of TMPRSS2-expressing Vero E6 cells. After 72 hr, cells were fixed and stained with crystal violet stain (1% wt/vol in 20% ethanol) allowing for the enumeration of viral plaques. The percent plaque inhibition for each nanobody dilution, summarized in *Figure 4— source data 1*, was used to fit the neutralization curves depicted in the figure. The colored shaded areas denote 90% confidence intervals for each fitted curve. $n \geq 3$.

The online version of this article includes the following source data for figure 4:

**Source data 1.** Neutralization data from the plaque reduction neutralization test (PRNT) assay with omicron BA.5 virus.

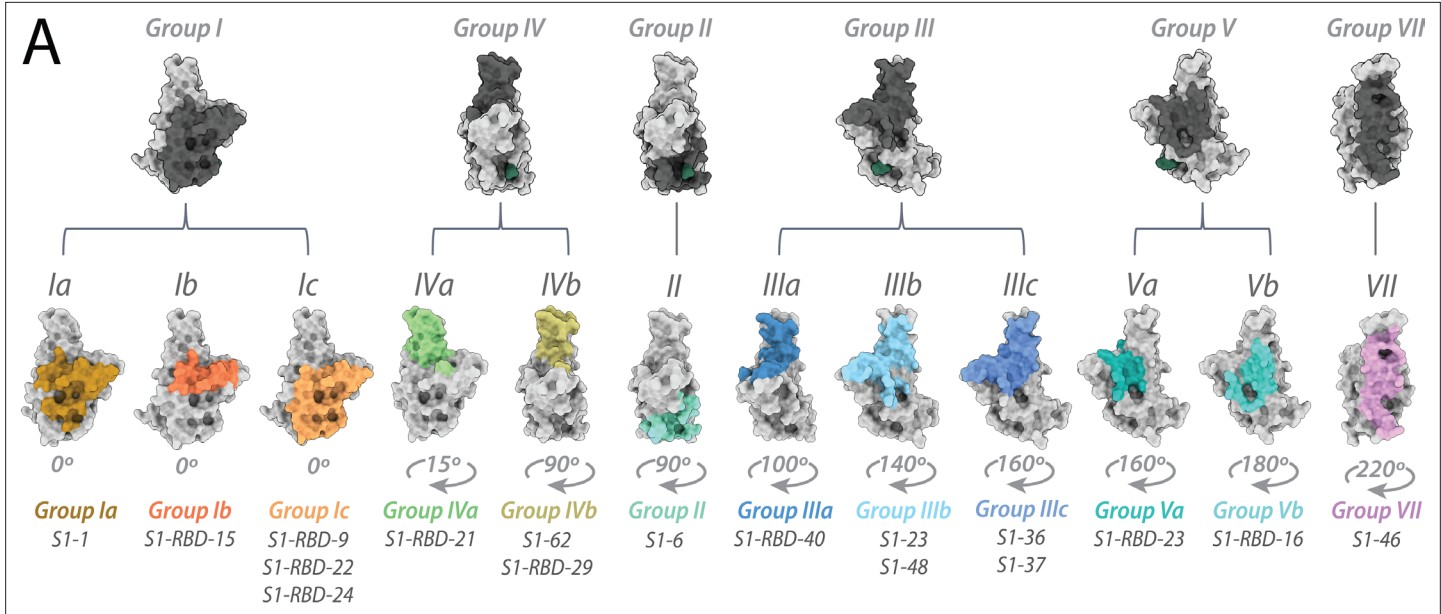

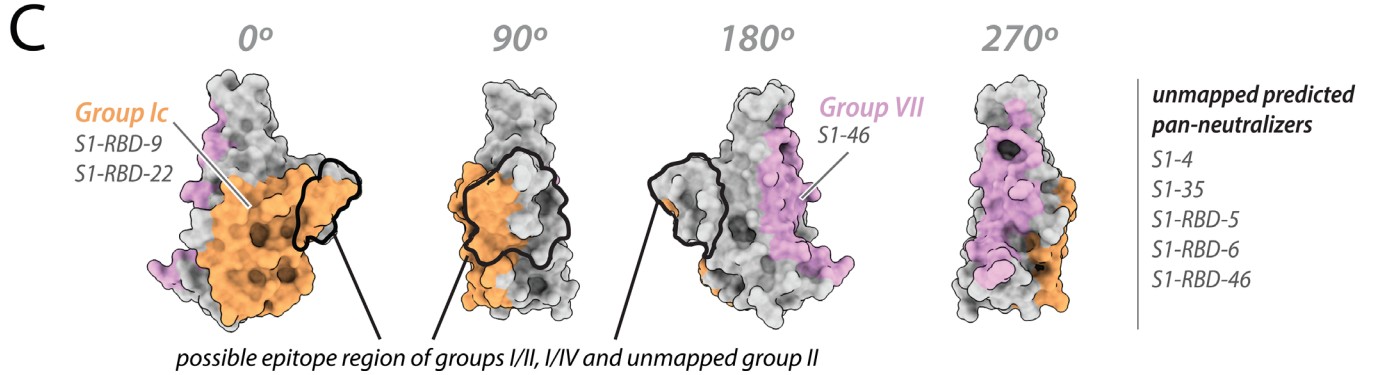

**Figure 5.** Refining epitopes of the nanobody repertoire. All epitopes are mapped on the structure of wild-type receptor-binding domain (RBD) (PDB ID: 6M0J). (**A**) The original six RBD nanobody epitope groups capable of binding and/or neutralizing one or more variants are highlighted in dark gray. Further refinement of four groups: I, III, IV, and V led to the identification of six additional epitope groups – resulting in a total of 12 epitope groups able to bind one or more variants of concern. (**B**) Summary of variant specific and broadly reactive nanobodies. (**C**) Nanobody groups predicted to bind/neutralize the circulating omicron variants EG.5 and HV.1. All structure representations were created on ChimeraX (*Pettersen et al., 2021*).

The online version of this article includes the following figure supplement(s) for figure 5:

**Figure supplement 1.** Mapping of variant-specific amino acid changes on different SARS-CoV-2 variants.

class 1 binders; groups II, III, IV, and V overlap with class 2 binders; and groups II, III, and V overlap with class 3 binders (*Figure 2*).

Interestingly, the epitope footprint of many of our nanobody classes extends beyond those of the three mAb classes (*Figure 2*). For example, the majority of the epitope regions of the group I, II, V, and VII nanobodies do not overlap with the mAb classes and are not a binding/neutralizing hotspot for mAbs (*Almagro et al., 2022*; *Cao et al., 2022*); instead, these epitopes extend away from the

ACE2-binding site (*Figures 1B, 2 and 3B*), as seen in particular with groups I, II, and V nanobodies. These regions may be inaccessible to mAbs, possibly due to steric limitations, a property nanobodies readily overcome due to their small size. Additionally, our data allowed us to identify variant-specific epitope groups (*Figure 5B*), where we define variant-specific nanobodies as nanobodies that bind a single additional variant alongside the original Wuhan strain, summarized as follows: of the 26 nanobodies that showed binding/neutralization to delta, 5 were specific only for delta; of the 21 nanobodies that showed binding/neutralization to omicron BA.1, 2 were specific for omicron BA.1. Strikingly, we have in our cohort eight nanobodies able to bind delta and the omicron lineages BA.1/ BA.4/BA.5/XBB/BQ.1.1 (*Figure 5B*). We further predict these eight nanobodies will be effective binders against current circulating strains of the virus including omicron EG.5 and HV.1 as the epitope regions (or predicted epitopes) of these nanobodies do not vary significantly from omicron lineages XBB and BQ.1.1 (*Figure 5C*; *Figure 5—figure supplement 1*).

## Nanobody synergy involving a non-neutralizing nanobody

We previously established that cocktails of nanobodies exhibit enhanced resistance to mutational escape (*Mast et al., 2021*). Excitingly, not only was the barrier to mutational escape extremely enhanced, but for certain combinations of nanobodies, their mechanisms of neutralization were synergistic, providing far more potent neutralization in combination than expected from the neutralization by either nanobody alone (*Mast et al., 2021*). Our present observations of nanobodies that retained binding to variants of the spike RBDdespite losing neutralization efficacy (*Figure 1A*) afforded us an opportunity to test whether the synergy observed for certain nanobody combinations was dependent on their ability to neutralize.

The synergistic S1-1 and S1-23 pair effectively neutralized the wild-type PSV with their epitopes on opposing surfaces of the RBD, permitting simultaneous binding and enhanced neutralization when delivered as a cocktail (*Mast et al., 2021*; *Figures 1A, 3A, and 6A, panel i*). While S1-1 remained efficacious against the delta variant of SARS-CoV-2, the L452R mutation in the delta RBD likely negatively impacted S1-23 (discussed above), weakening its binding affinity by ~1000-fold (12 nM) and negating its neutralization efficacy at concentrations <10 μM (*Figures 1A, 3A, and 6A, panel ii*). Surprisingly, when provided in combination with S1-1, which displays increased binding affinity and enhanced neutralization against the delta variant, S1-23 was able to further enhance the neutralization capabilities of S1-1, synergistically, at concentrations above $10^{-3}$ μM (*Figure 6A, panel ii*), by up to 42-fold. This synergistic interaction, however, did not apply to situations where extensive mutations are present in the RBD, such as in the omicron sublineages of SARS-CoV-2, which ablated the binding and neutralization efficacy of both S1-1 and S1-23 (*Figures 1A, 3A, and 6A, panel iii*).

We also tested the broadly neutralizing nanobody S1-RBD-22 in combination with S1-36 (*Figure 6B*). Like S1-23, the epitope of S1-36 is opposite that of S1-RBD-22, permitting simultaneous binding to a single RBD (*Mast et al., 2021*). However, while its neutralization efficacy dropped off when delivered to either delta or omicron BA.1 PSV alone, its ability to bind to its epitope was only marginally impacted (*Figures 1A, 3A, and 6B*). When provided in combination with S1-RBD-22, S1-36 synergistically enhanced by up to 80-fold the neutralization efficacy of S1-RBD-22 against both delta and omicron BA.1 PSV (*Figure 6B, panels ii, iii*). The synergies observed between S1-1 and S1-23, and between S1-RBD-22 and S1-36, appear to be specific rather than due to pleiotropic effects. This is evidenced by the lack of enhanced neutralization when S1-1 is combined with the non-specific LaM2 nanobody against delta PSV (*Figure 6C*). Furthermore, in the case of S1-23, binding to its non-neutralizing epitope on the RBD of delta PSV was able to induce dose-dependent antagonistic effects on the neutralizing efficacy of S1-RBD-16, which binds to a neighboring epitope that can be competitively blocked by S1-23 (*Figure 6D*; *Mast et al., 2021*).

## Conclusions

Collectively, our binding and neutralization data allowed us to identify the regions of spike in multiple VoC that remained vulnerable to our original repertoire of nanobodies raised against wild-type SARS-CoV-2 spike (*Figure 7*; *Mast et al., 2021*). Unlike the epitope groups defining mAb-binding sites on spike, many of our nanobody epitope groups remained efficacious in the neutralization of different VoC (*Figures 6 and 7*), possibly due to each nanobody's smaller epitope footprint allowing their access to regions of spike inaccessible to mAbs. However, the substantial changes on the surface of

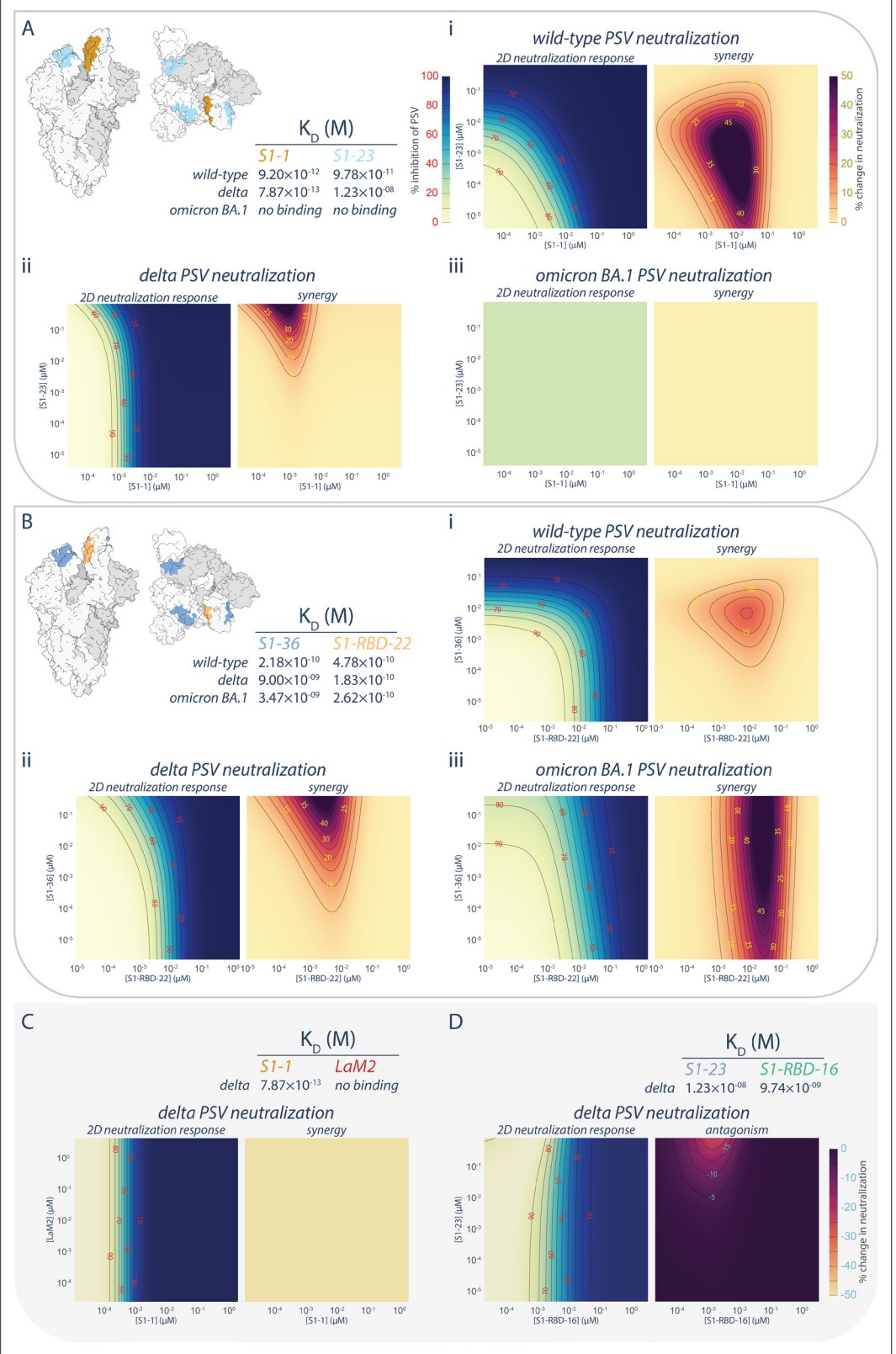

**Figure 6.** Persistence of synergistic neutralization with nanobody cocktails against SARS-CoV-2 variants of concern. (**A**) S1-1 synergizes with S1-23 in neutralizing SARS-CoV-2 PSV. The upper left panel shows two representations of spike with the accessible S1-1 (dark goldenrod) and S1-23 (sky blue) epitopes (PDB ID: 6VYB). The measured affinities for S1-1 and S1-23for the receptor-binding domain (RBD) of wild-type, delta, and omicron BA.1 are

*Figure 6 continued on next page*

*Figure 6 continued*

displayed. Both S1-1 and S1-23 neutralize wild-type (**i**), whereas only S1-1 neutralizes delta at the concentrations shown (**ii**). In spite of a lack of neutralization at these concentrations, S1-23, synergizes with S1-1 and enhances its neutralization of delta SARS-CoV-2 PSV (**ii**). As neither S1-1 nor S1-23 are able to bind to the RBD of omicron BA.1, neither nanobody neutralizes omicron BA.1 SARS-CoV-2 PSV (**iii**). (**B**) S1-36 synergizes with S1-RBD-22 in neutralizing SARS-CoV-2 PSV. As in A, the upper left panel shows two representations of spike with the accessible S1-36 (cornflower blue) and S1-RBD-22 (sandy brown) epitopes. The measured affinities for S1-36 and S1-RBD-22 are displayed. Both S1-36 and S1-RBD-22 neutralize wild-type (**i**), whereas only S1-RBD-22 effectively neutralizes delta and omicron BA.1 SARS-CoV-2 PSV at the concentrations shown (**ii** and **iii**, respectively). However, S1-36 synergizes with S1-RBD-22 and enhances its neutralization of the three depicted SARS-CoV-2 pseudoviruses (**i**, **ii**, and **iii**). (**C**) An example of no interactions (synergistic or antagonistic) between S1-1 and LaM2 (*Fridy et al., 2014*), a non-specific nanobody that does not bind the RBD of delta. (**D**) An example of antagonism, where higher concentrations of S1-23 interferes with the ability of S1-RBD-16 to neutralize delta SARS-CoV-2 PSV. These nanobodies have adjacent epitopes on the RBD of spike and were previously shown to interfere with each other's binding to their respective epitope (*Mast et al., 2021*). *n* = 4. Source data in .

The online version of this article includes the following source data for figure 6:

**Source data 1.** Neutralization data from synergy experiment.

___

spike that has occurred as SARS-CoV-2 has evolved from one variant to the next has also negatively impacted many nanobodies by abolishing or weakening their binding and/or neutralization activity. This weakening is most evident for nanobodies directed against the NTD, and against the receptor-binding motif that engages ACE2 (*Figures 1B and 2B*).

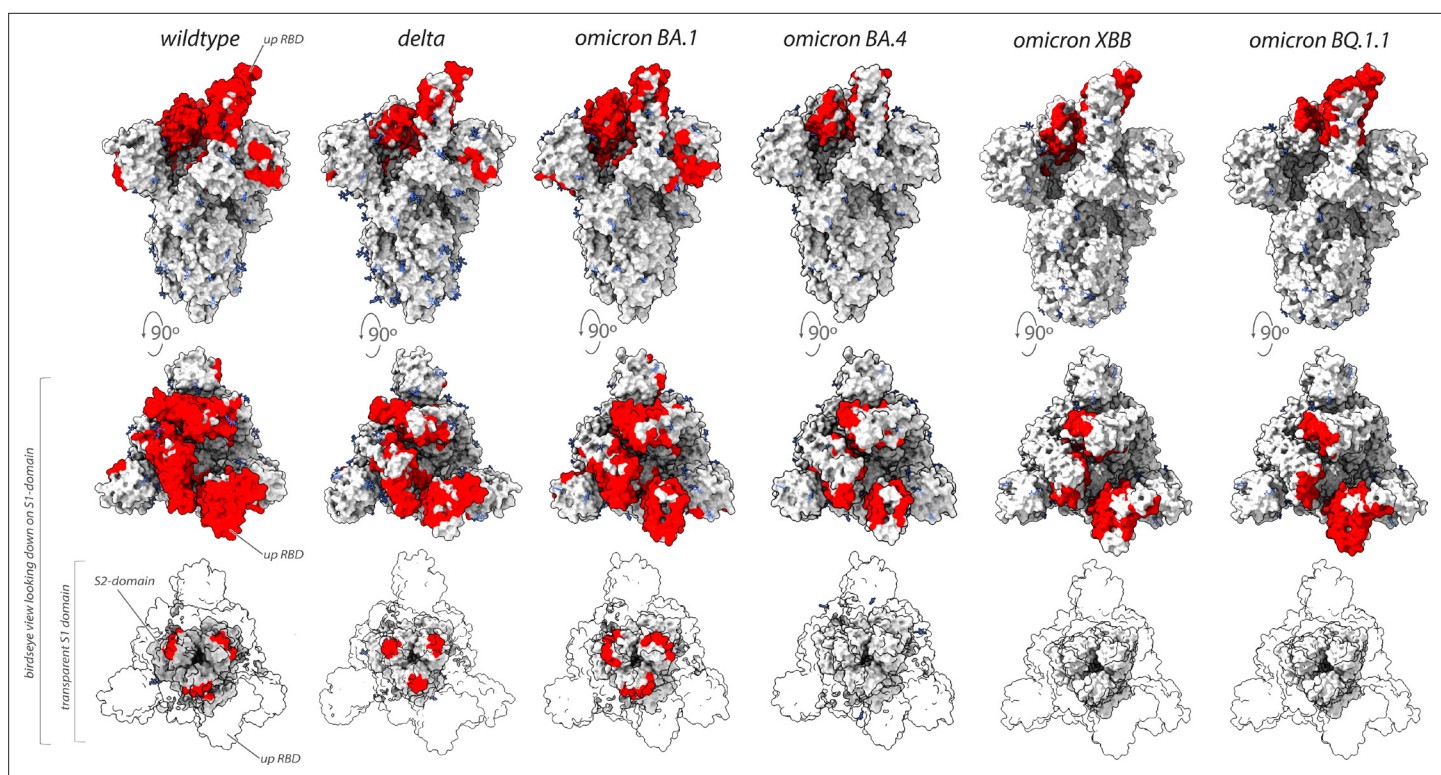

**Figure 7.** Nanobodies effective against circulation variants of concern. The nanobody epitopes (red) that retain effectiveness against wild-type (PDB ID: 7KNB), delta (PDB ID: 7V7O), omicron BA.1, omicron BA.4 (the nanobody epitopes were mapped on to PDB ID: 7XO5 for both BA.1 and BA.4 due to the lack of a suitable BA.4 structure),omicron XBB.1 and BQ.1.1 (the nanobody epitopes were mapped on to PDB ID: 8IOU for both XBB.1 and BQ.1.1 due to the lack of a suitable BQ.1.1 structure). The spike trimer (*silver*) where glycans are represented as *light blue* sticks for each variant is displayed in three views: top = side view of spike trimer; middle = birds eye view looking down on the S1 domain and; bottom = birds eye view (same as middle view), with the S1 domain rendered transparent to enable visualization of the S2 domain. All structure representations were created on ChimeraX (*Pettersen et al., 2021*).

Intriguingly, we discovered nanobodies that maintained binding capabilities while losing their neutralizing properties. This finding paves the way for engineering these nanobodies through approaches like oligomerization, which has proven effective in boosting neutralization (*Wrapp et al., 2020*; *Mast et al., 2021*). Remarkably, we also showed that such non-neutralizing binders can nevertheless retain effectiveness as components of synergistic nanobody cocktails, suggesting their potential for widespread antiviral applications. This finding indicates that the mechanism of synergy can operate through epistatic interactions from binding alone, not solely through direct neutralization. Furthermore, these synergistic pairings hold promise as therapeutic options when formulated as multivalent hetero-oligomers and we propose that any of the nanobodies we have demonstrated to show pan-VoC activity would be prime candidates for further optimization.

In this study, we identified nanobodies that specifically recognize only certain SARS-CoV-2 variants (*Figure 5B*), allowing for the possibility of distinguishing between different VoC. By utilizing these nanobodies as molecular probes in diagnostic tests, a unique 'molecular fingerprint' could define each variant based on the combinations of nanobodies that bind to and recognize the specific virus particle. Consequently, this approach could enable the accurate and rapid diagnosis of SARS-CoV-2 infections as well as provide real-time identification of the specific variant causing the infection, thus enhancing surveillance and tailoring treatment strategies accordingly to optimize patient outcomes and control the spread of the virus.

Together, our findings highlight the strength and variety of the heavy chain-only category of llama antibody immune responses and, as a result, the extensive repertoires of high-affinity nanobodies. This result emphasizes the distinct benefits of nanobody technology, which enables extensive coverage of antigenic regions while simultaneously targeting multiple unique epitope sites. This approach paves the way for investigating a wide range of therapeutic options against rapidly evolving proteins like the SARS-CoV-2 spike protein, ultimately aiding in our preparedness and defense against future pandemics or major outbreaks.

# Materials and methods

## $K_D$ measurements

Series S CM5 sensor chips (Cytiva) were immobilized with recombinant delta Spike S1, delta RBD, and omicron RBD at 12.5, 10, and 15 µg/ml, respectively, using EDC/NHS coupling chemistry according to the manufacturer's instructions. $K_D$ measurements were performed on a Biacore 8k (Cytiva) as previously described (*Mast et al., 2021*), using 5–8 concentrations of each nanobody. Data were processed and analyzed using the Biacore Insight Evaluation software.

## Structural modeling

Integrative structure modeling of nanobody epitopes on the different variant spike proteins, proceeded through the standard four-stage protocol (*Kim et al., 2018*; *Webb et al., 2018*; *Rout and Sali, 2019*; *Sali, 2021*; *Saltzberg et al., 2021*). This protocol was implemented using the *Python Modeling Interface* package, a library for modeling macromolecular complexes based on the open-source *Integrative Modeling Platform* software, version 2.15.0 (https://integrativemodeling.org). Only (a subset of) nanobodies with pre-determined experimental escape mutations on the Wuhan spike structure were selected for modeling. Separate models were computed for rigid-receptor–rigid ligand-type binary docking of representative nanobodies from group 1 (S1-1, S1-RBD-[9, 15, 22, 24]), group 2 (S1-6), group 3 (S1-[23, 36]), group 4 (S1-RBD-[21, 29]), group 5 (S1-RBD-[16, 23]), group 7 (S1-46), group 8 (S1-49), and group 9 (S2-10), to the variant spike structures. S1-49 was docked to a monomeric S1-NTD domain, S2-10 was docked to the trimeric S2 (ecto-) domain, while the remaining nanobodies were docked to the S1-RBD domain. Monomeric S1-RBD spanning amino acid residues 333–526 was represented using the 2.45 Å crystal structure of the ACE2-bound RBD (6M0J.E; *Lan et al., 2020*) for the original virus, the 4.30 Å cryo-EM structure of up-RBD pre-fusion spike (7SBO.A; *Zhang et al., 2021*) for the delta variant, the ACE2 bound 2.45 Å cryo-EM spike structure (7T9K.B; *Mannar et al., 2022*) for the omicron BA.1 variant and a structure predicted using AlphaFold-2 for the omicron BA.4/BA.5 variant, the 3.18 Å cryo-EM structure of ACE2-bound up-RBD pre-fusion spike (8IOU.A; *Tamura et al., 2023*) for the XBB.1 variant, and the 3.20 Å cryo-EM structure of the BQ.1.1

variant RBD (8FXC:E; *Tamura et al., 2023*) in complex with ACE2 and the S309-neutralizing antibody Fab fragment. Missing residues in 8FXC:E were filled in using Modeller (*Webb and Sali, 2016*).

For the original virus, monomeric S1-NTD, spanning amino acid residues 16–305, was represented using the crystal structure of the S2M28 Fab bound NTD (7LY3.A; *McCallum et al., 2021*), while trimeric S2 was represented using the amino residues 689–1162 (for each monomer) from the 2.9 Å cryo-EM structure 6XR8 (*Cai et al., 2020*). NTD and S2 structures for delta and omicron variants were extracted from the corresponding whole spike structures.

Structural models for all 15 nanobodies and the omicron BA.4 RBD were built with the ColabFold implementation of AlphaFold2 (*Jumper et al., 2021*; *Mirdita et al., 2022*). The protocol included automatic refinement of the complementarity-determining region (CDR) loops through an all-atom energy minimization of the AlphaFold2-predicted structure using the AMBER molecular mechanics force field (*Hornak et al., 2006*). We verified that the predicted nanobody structures are within 3–4 Å backbone root-mean-square deviation (RMSD) from the comparative models of these nanobody sequences published previously (*Mast et al., 2021*). The CDR region boundaries in the nanobody structures were assigned using the FREAD algorithm as implemented within the SabPred web server (*Rausch, 1991*).

To make structural sampling sufficiently efficient, the system was represented at a resolution of one bead per residue, and the receptors and all nanobodies were treated as rigid bodies. For each nanobody, alternate binding modes were scored using spatial restraints enforcing receptor–ligand shape complementarity, cross-link satisfaction and proximity of CDR3 loops on the nanobodies to escape mutant residues on the corresponding receptor. With the receptor fixed in space, 1,200,000 alternative docked nanobody models were produced through 20 independent runs of replica exchange Gibbs sampling based on the Metropolis Monte Carlo algorithm, where each Monte Carlo step consisted of a series of random rotations and translations of rigid nanobodies. The initial set of models was filtered to obtain a random subsample of 30,000 models, which were clustered by the structural similarity of their interfaces to the receptor; this similarity was quantified by the fraction of common contacts (fcc) between receptor and nanobody was used to characterize interface similarity between alternate nanobody poses (*Rodrigues et al., 2012*). Binding poses from the most populated cluster were selected for further analysis. Five independent random subsamples of 30,000 models each were generated from the set of all models post-structural sampling, and the entire protocol of interface similarity-based clustering and top cluster selection was repeated each time. No significant differences among these five subsamples were observed in the satisfaction of restraints. Structural differences among the variants, as well as between AlphaFold2 models and previously published comparative models of nanobodies, lead to differences in binding modes of the same nanobody to different spike variants. Thus, for the sake of consistency we limit our comparison to the receptor epitopes, which are defined as all receptor atoms that are within 6 Å of the framework and CDR regions of the nanobodies (excluding the flexible N- and C-terminal regions). Although we do not include the nanobody paratopes in our analysis, we verified that all binding modes are primarily through CDR3, except for the CDR1 contribution to the binding of S1-1 and S1-RBD-15 to the RBD, for all variants. Relative differences in binding affinity ($K_D$) and neutralization potential ($IC_{50}$) between the original virus and other variants (delta and omicron) were projected onto the Wuhan epitopes to create the heatmaps in *Figures 1B and 2B*. Relative differences were reported as and/or normalized to the range from 0 to 100.

Integrative models of nanobody epitopes on the spike protein were computed on the Wynton HPC cluster at UCSF. Receptor epitopes were visualized in UCSF ChimeraX (*Pettersen et al., 2021*). Files containing input data, scripts, and output results are available at https://github.com/integrative-modeling/nbspike/tree/main/integrative_modeling_VOC (copy archived at *Sanyal, 2024*). Structure predictions using ColabFold utilized the 'AlphaFold2_batch' notebook, with the default settings. All modeled structures were subjected to molecular-mechanics-based relaxation, followed by using the model with the top pLDDT score was selected for the integrative modeling pipeline.

## Cell lines

TMPRSS2-expressing Vero E6 cells, 293T/17 cells and 293T-hACE2 cells were cultured as described previously (*Mast et al., 2021*). Briefly, TMPRSS2 + Vero E6 cells were cultured at 37°C in the presence of 5% $CO_2$ in medium composed of in high-glucose Dulbecco's modified Eagle's medium (DMEM, Gibco) supplemented with 10% (vol/vol) fetal bovine serum (FBS) and 1 mg/ml geneticin. 293T/17

were cultured at 37°C in the presence of 5% $CO_2$ in a medium composed of DMEM supplemented with 10% (vol/vol) FBS and penicillin/streptomycin. 293T-hACE2 cells were cultured at 37°C in the presence of 5% $CO_2$ in medium composed of DMEM supplemented with 10% (vol/vol) FBS, penicillin/streptomycin, 10 mM HEPES (4-(2-hydroxyethyl)-1-piperazineethanesulfonic acid), and with 0.1 mM modified Eagle's medium (MEM) non-essential amino acids (Thermo Fisher). All experiments were performed with cells passaged less than 15 times. The identities of cell lines were confirmed by chromosomal marker analysis and tested negative for mycoplasma using a MycoStrip (InvivoGen).

## Production of SARS-CoV-2 variant pseudotyped lentiviral reporter particles

Pseudovirus stocks were prepared and ittered as described previously (*Mast et al., 2021*). Variant spike containing plasmids were combined with pHAGE-CMV-Luc2-IRES-ZsGreen-W (BEI Cat # NR-52516) (*Crawford et al., 2020*), and psPAX using lipofectamine 3000 and cotransfected into 293T/17 cells. Pseudovirus was titered by threefold serial dilution on 293T-hACE2 cells, as described previously (*Mast et al., 2021*).

## SARS-CoV-2 pseudovirus neutralization assay

Nanobodies were tested for their neutralization properties as described previously (*Mast et al., 2021*). Briefly, threefold serial dilutions of nanobodies were incubated with pseudotyped SARS-CoV-2 for 1 hr at 37°C. The nanobody–pseudovirus mixtures were then added in quadruplicate to 293T-hACE2 cells along with 2 µg/ml polybrene (Sigma). Cells were incubated at 37°C with 5% $CO_2$. Infected cells were processed between 52 and 60 hr by adding equal volume of Steady-Glo (Promega), and firefly luciferase signal was measured using the Biotek Model N4 with integration at 0.5 ms. Data were processed using Prism 7 (GraphPad), using four-parameter non-linear regression (least-squares regression method without weighting). All nanobodies were tested at least two times and with more than one pseudovirus preparation.

## Plaque reduction neutralization assay with SARS-CoV-2 BA.5

Briefly, 10 threefold serial dilutions of representative nanobodies of select epitope groups in Opti-MEM (Gibco) were incubated with approximately 100–200 pfus of SARS-CoV-2 BA.5 for 1 hr at RT. The nanobody/virus mixture was then added to a confluent monolayer of TMPRSS2 + Vero E6 cells in 12-well plates and incubated at RT for 90 min. One well was overlaid with virus only while another well was uninfected. Virus/nanobody mixture was removed, and the cell monolayer overlaid with a medium composed of 3% (wt/vol) carboxymethycellulose and 4% (vol/vol) FBS in Opti-MEM. 96 hr post infection, the overlay was removed and cell monolayer was washed with Dulbecco's phosphate-buffered saline (DPBS, Gibco) before being fixed with 4% (wt/vol) paraformaldehyde in DPBS for 30 min. Fixative was removed and the cells were rinsed with DPBS before being stained with 1% (wt/vol) crystal violet in 20% (vol/vol) ethanol. Contrast was enhanced by washing with DPBS, and clear plaques representing individual viral infections were visualized as spots lacking crystal violet stain. Plaques were quantified and the ratio of plaques at each nanobody dilution to 'virus only' well was used to determine the $IC_{50}$s of each nanobody.

## Nanobody synergy

Synergy experiments were performed as described previously (*Mast et al., 2021*). Briefly, a robotic liquid handler was used to prepare 2D matrices of threefold serial dilutions of two nanobodies and then mix these combinations with different variant pseudotyped SARS-CoV-2 for 1 hr. After incubation with the virus, the mixture was overlaid on a monolayer of 293-hACE2 cells and left to incubate for 56 hr. Luminescence was quantified as described above. Data were processed using the Bivariate Response to Additive Interacting Doses (BRAID) model (*Twarog et al., 2016*) as implemented in the synergy software package for python (*Wooten and Albert, 2021*).

## SARS-CoV-2 stocks and titers

All experimental work involving live SARS-CoV-2 was performed at Seattle Children's Research Institute (SCRI) in compliance with SCRI guidelines for BioSafety Level 3 (BSL-3) containment. SARS-CoV-2 isolate CGIDR_SARS2 omicron BA.5 was obtained from an infected individual. An initial inoculum

was diluted in Opti-MEM (Gibco) at 1:1000, overlaid on a monolayer of Vero E6 and incubated for 90 min. Following the incubation, the supernatant was removed and replaced with 2% (vol/vol) FBS in Opti-MEM medium. The cultures were inspected for cytopathic effects, and infectious supernatants were collected, cleared of cellular debris by centrifugation, and stored at −80°C until use. Whole viral genome sequencing and variant analysis were performed by the University of Washington Department of Laboratory Medicine & Pathology. Viral titers were determined by plaque assay using a liquid overlay and fixation-staining method, as described previously (*Mendoza et al., 2020*; *Mast et al., 2021*).

## Acknowledgements

We are very grateful to The Fisher Drug Discovery Resource Center (DDRC), Rockefeller University (RRID:SCR_020985), and the rest of the Aitchison, Chait and Rout laboratories for intellectual support. Funding. American Lung Association (JDA, BTC, MPR), G Harold and Leila Y Mathers Charitable Foundation (JDA, BTC, MPR), Robertson Therapeutic Development Fund (JDA, BTC, MPR) Jain Foundation (JDA, BTC, MPR), National Institutes of Health grant P41GM109824 (JDA, BTC, AS, MPR), and National Institutes of Health grant R01GM083960 (AS).

## Additional information

### Competing interests

Natalia E Ketaren, Fred D Mast, Peter C Fridy, Jean Paul Olivier, Brian T Chait, Michael P Rout, John D Aitchison: Inventor on a provisional patent (US20230331824A1) describing the anti-spike nanobodies described in this manuscript. The other authors declare that no competing interests exist.

### Funding

| Funder | Grant reference number | Author |
|---|---|---|
| G. Harold and Leila Y. Mathers Foundation | | Brian T Chait Michael P Rout John D Aitchison |
| Robertson Therapeutic Development Fund | | Brian T Chait Michael P Rout John D Aitchison |
| Jain Foundation | | Brian T Chait Michael P Rout John D Aitchison |
| National Institutes of Health | P41GM109824 | Andrej Sali Brian T Chait Michael P Rout John D Aitchison |
| National Institutes of Health | R01GM083960 | Andrej Sali |
| American Lung Association | 923055 | Brian T Chait Michael P Rout John D Aitchison |

The funders had no role in study design, data collection, and interpretation, or the decision to submit the work for publication.

### Author contributions

Natalia E Ketaren, Fred D Mast, Peter C Fridy, Jean Paul Olivier, Tanmoy Sanyal, Conceptualization, Formal analysis, Validation, Investigation, Visualization, Methodology, Writing – original draft, Writing – review and editing; Andrej Sali, Supervision, Funding acquisition, Writing – review and editing; Brian T Chait, Michael P Rout, John D Aitchison, Conceptualization, Supervision, Visualization, Methodology, Project administration, Writing – review and editing

## Author ORCIDs

Natalia E Ketaren ⓘ https://orcid.org/0000-0002-7869-6162
Fred D Mast ⓘ https://orcid.org/0000-0002-2177-6647
Peter C Fridy ⓘ https://orcid.org/0000-0002-8208-9154
Jean Paul Olivier ⓘ https://orcid.org/0000-0002-2197-271X
Tanmoy Sanyal ⓘ http://orcid.org/0000-0002-6009-9431
Andrej Sali ⓘ http://orcid.org/0000-0003-0435-6197
Michael P Rout ⓘ https://orcid.org/0000-0003-2010-706X
John D Aitchison ⓘ https://orcid.org/0000-0002-9153-6497

Reviewer #1 (Public Review): https://doi.org/10.7554/eLife.89423.3.sa1
Reviewer #2 (Public Review): https://doi.org/10.7554/eLife.89423.3.sa2
Author response https://doi.org/10.7554/eLife.89423.3.sa3

---

# Additional files

## Supplementary files

• MDAR checklist

## Data availability

All data generated or analyzed during this study are included in the manuscript and supporting files.

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
