## [Editor Report · eLife assessment]

This study presents **important** insights on the impact of SARS-CoV-2 variants on the binding and neutralization of a small library of nanobodies. The authors should be applauded for their comprehensive in vitro and in silico analyses of nanobody targeting of SARS-CoV-2 variants. The evidence supporting the claims of the authors is now **convincing**. This work will be of great interest to researchers in the fields of antibody/nanobody engineering and SARS-CoV-2 therapeutics.

---

## [Referee Report · Reviewer #1 (Public Review)]

Summary:

In this manuscript, Ketaren, Mast, Fridy et al. assessed the ability of a previously generated llama nanobody library (Mast, Fridy et al. 2021) to bind and neutralize SARS-CoV-2 delta and omicron variants. The authors identified multiple nanobodies that retain neutralizing and/or binding capacity against delta, BA.1 and BA.4/5. Nanobody epitope mapping on spike proteins using structural modeling revealed possible mechanisms of immune evasion by viral variants as well as mechanisms of cross-variant neutralization by nanobodies. The authors additionally identified two nanobody pairs involving non-neutralizing nanobodies that exhibited synergy in neutralization against the delta variant. These results enabled the refinement of target epitopes of the nanobody repertoire and the discovery of several pan-variant nanobodies for further preclinical development.

Strengths:

Overall, this study is well executed and provides a valuable framework for assessing the impact of emerging SARS-CoV-2 variants on nanobodies using a combination of in vitro biochemical and cellular assays as well as computational approaches. There are interesting insights generated from the epitope mapping analyses, which offer possible explanations for how delta and omicron variants escape nanobody responses, as well as how some nanobodies exhibit cross-variant neutralization capacity. These analyses laid out a clear path forward for optimizing these promising next-gen therapeutics, particularly in the face of rapidly emerging SARS-CoV-2 variants. This work will be of interest to researchers in the fields of antibody/nanobody engineering, SARS-CoV-2 therapeutics, and host-virus interaction.

Weaknesses:

A main weakness of the study is that the efficacy statement is not thoroughly supported. While the authors comprehensively characterized the neutralizing ability of nanobodies in vitro, there is no animal data involving mice or hamsters to demonstrate the real protective efficacy in vivo. Yet, in the title and throughout the manuscript, the authors repeatedly used phrases like "retains efficacy" or "remains efficacious" to describe the nanobodies' neutralization or binding capacities. This claim is not well supported by the data and underestimates the impact of variants on the nanobodies, especially the omicron sublineages. For example, the authors showed that S1-RBD-15 had a ~100-fold reduction in neutralization titer against Omicron, with an IC50 at around 1 uM. This is much higher than the IC50 value of a typical anti-ancestral RBD nanobody reported in the previous study (Mast, Fridy et al. 2021). In fact, the authors themselves ascribe nanobodies with an IC50 above 1 uM as weak neutralizers. And there were many in the range of 0.1-1 uM. Furthermore, many nanobodies selected for affinity measurement against BA.4/5 had no detectable binding. Without providing in vivo protection data or including monoclonal antibodies that are known to be efficacious against variants in the in vitro assays as a benchmark, it is difficult to evaluate the efficacy just with the IC50 values.

Comments post revision:

The authors are to be commended for their comprehensive response to the referees' comments. In the revised manuscript, the authors made extensive changes throughout the texts and added new figures that greatly improved their clarity. While the manuscript is still limited in solely relying on in vitro data for efficacy assessment, it nicely demonstrates how the combination of experimental and computational techniques could lead to the discovery of broadly neutralizing nanobody candidates for further lead optimization.

---

## [Referee Report · Reviewer #2 (Public Review)]

Summary:

Interest in using nanobodies for therapeutic interventions in infectious diseases is growing due to their ability to bind hidden or cryptic epitopes that are inaccessible to conventional immunoglobulins. In the presented study, authors posed to characterize nanobodies derived the library produced earlier with Wuhan strain of SARS-CoV-2, map their epitopes on SARS-CoV-2 spike protein and demonstrate that some nanobodies retain binding and even neutralization against antigenically distant, newly emerging Variants of Concern (VOCs).

Strengths:

Authors demonstrate that some nanobodies despite being obtained against ancestral virus strain retain high affinity binding to antigenically distant SARS-CoV-2 strains despite majority of the repertoire loses binding. Despite being limited to only two nanobody combinations, demonstration of synergy in virus neutralization between nanobodies targeting different epitopes is compelling. The ability of nanobodies to bind emerging virus strains has been demonstrated and the possible effect of mutations within epitopes has been thoroughly discussed.

---

## [Author Response]

The following is the authors’ response to the original reviews.

**Reviewer #1 (Recommendations For The Authors):**
Beyond my general review, some descriptions of the results and methods could be further clariﬁed, which I've outlined below:(1) Page 3, Line 118-120: Based on results from Fig 1A, the authors reported 15 nanobodies neutralized both delta and BA.1 out of the 41 tested. However, I only counted 14. Could the authors double check?

We recounted the nanobodies and confirmed there are 15 as follows:

(1) RBD-15

(2) RBD-22

(3) RBD-24

(4) RBD-9S1-4

(5) S1-35

(6) RBD-6

(7) RBD-5

(8) RBD-21

(9) RBD-16

(10) S1-46

(11) S1-49dimer

(12) S2-10dimer

(13) S2-3

(14) S2-62

(2) Page 5, Lines 134-135: the authors described that the heatmap reflects the neutralizing strength of the representative nanobodies from each group. For groups where multiple nanobodies were selected for visualization, how was the neutralization strength calculated? Was the IC50 averaged first before being converted into the neutralization strength?

This has been made clear in the legend for Fig. 1 as follows “For groups with multiple nanobodies, the average -log10 (IC50) is first calculated for the nanobodies within that group, then normalized to a neutralization score within the 0–100 range using the min and max average -log10 (IC50) for that group. A higher score indicates more potent neutralization of the variant relative to the wild type.”

(3) Page 5, Lines 138-139: What was the authors' rationale for selecting certain nanobodies over others for structural modeling and visualizing the neutralization heatmap in Fig 1B? Does it introduce bias to the neutralizing epitope map on the spike protein?

We only focused on nanobodies for which we had enough epitope mapping data to unambiguously generate docked nanobody-spike models, as explained in our previous study (Mast et. al, eLife 2021). When multiple nanobodies within the same group had sufficient epitope mapping data available, we selected only representative candidates that had better binding affinity and/or neutralization potency. As epitope mapping via escape mutants relied largely on random point mutagenesis of Spike, there should be little introduced bias.

Overall, groups I-VII cover an exhaustive set of target areas on the RBD (including the lone glycan site in Group-II), while groups VII and IX are representative areas on NTD and S2. Using group-average IC50s and suitable normalization as mentioned in point 3 above further prevent potential biases due to unequal number of Nbs modeled from each group.

We have modified the text with the following:

“For computational epitope modeling, we selected nanobody candidates using a series of experimentally obtained structural restraints, as described in Mast, Fridy et al. 2021.”

(4) Page 5, Lines 161-167: It would be good to include Fig S1 as a main figure as it places the epitope landscape of nanobodies being investigated in this manuscript into the broader context of clinically approved monoclonal antibody therapeutics for COVID-19.

We have amended the Figures to accommodate the reviewers suggestion. Figure S1 is now Figure 2.

(5) Page 6, Lines 173-175: The neutralization breadth for S1-46 is quite encouraging. Any speculations on why this particular nanobody is so broadly targeting? Any additional thoughts on why its high binding affinity (nM) did not translate into strong neutralization (as it is in the 0.1-1 uM range)?

S1-46 binds a region on spike that is conserved across all variants observed to date. Its epitope is difficult to access unless the RBD is in the up conformation, which may explain why monoclonal antibodies rarely bind. We state this in the text as follows:

“S1-46 binds a region on spike that is conserved across all variants to date, but which may be relatively inaccessible and is not targeted by any of the mAbs that previously received EUA by the FDA (Cox, Peacock et al. 2023).”

Relating neutralization activity to binding activity requires more insight into the mechanisms of binding and activity. Nonetheless, we are also encouraged by S1-46’s breadth and numerous avenues can be pursued to greatly improve its neutralizing activity (e.g. synergistic combinations).

(6) Page 6, Lines 173-175: For the remaining two nanobodies S1-31 and S1-RBD-11 in group VII, the target epitopes on the spike proteins of either delta or BA.1 do not seem to bear any mutations, at least based on the mutation maps in Fig 1B. Yet their neutralizing capacities against delta and BA.1 variants were abolished. Do the authors have any idea about what is going on here?

For group VII, only the epitope of S1-46 was mapped whereas S1-31 and S1-RBD-11 were assigned to group VII based on our lower resolution binning experiments. Thus, without knowing precisely where they bind, we can make only limited conclusions at this time. In the absence of supporting structural information, we speculate that the epitopes of RBD-11 and S1-31 may be in a region that overlaps with or is in close proximity to a mutation that could affect the binding of the nanobody enough to result in loss of neutralizing ability.

(7) Page 7, Line 195-200: Please provide PRNT50 or logPRNT50 for the five nanobodies selected for BA.4/5 PRNT assay.

We have added this suggested information. Additionally, a supporting table (Table S1) is now provided.

(8) Page 8, Lines 223-224: Similar to comment 3, what was the rationale here for choosing certain nanobodies over others for structural modeling and visualizing the binding heatmap in Fig 2B?

The set of nanobodies chosen for structural modeling and visualization of neutralization data is identical to the set of anti-RBD nanobodies chosen for binding.

(9) Page 11, Lines 326-328: Can the authors include mutation maps as part of Fig 4C to show the mutation distributions on the XBB/BQ.1/BQ/1.1 spikes?

We have updated and added a supplemental figure to accompany Fig. 5 (called “supplement for Figure 5”) showing the mutation maps.

(10) Page 14, Line 409-418: This paragraph is well considered. Given the large number of nanobodies assessed in this manuscript, it would be helpful if the authors could highlight some candidate nanobodies as lead candidates for further optimization.

While our intention in this manuscript was not to provide targeted recommendations for lead candidates, but rather to reiterate the collective potential of a Nb pool originally targeted towards the 2019 Wuhan variant, the reviewers point is interesting. We speculate that any of the Nbs we have demonstrated to show pan-VoC activity, would be prime candidates for further optimization.

We have added a statement to this effect as follows: “We propose that any of the Nbs we have demonstrated to show pan-VoC activity, would be prime candidates for further optimization.”

**Reviewer #2 (Recommendations For The Authors):**
Major concerns:(1) The main message of the article is the prediction that nanobodies that retain binding to the different SARS-CoV-2 variants including early Omicron strains will retain binding and neutralization against currently circulating strains such XBB and BQ. However, no evidence either via modeling or experimental testing has been provided for that prediction. The study will benefit from mapping amino acid mutations in RBD of XBB and BQ lineages compared to BA.4/5 and demonstrating via computation docking that epitopes of the five nanobodies that retain binding to BA.4/5 RBD are not affected. For example, the crystal structure of XBB.1 RBD PDB:8OIV is available. Binding/neutralization experiment with currently circulating SARS-CoV-2 strains would still be the gold standard test given the fact that only five out of 41 nanobodies retained binding and neutralization to BA.4/5 lineage. Loss of neutralization ability against BA.4/5 without a significant decrease in binding affinity for nanobodies S1-46 and S1-RBD-22 further indicates that neutralization of XBB and BQ lineage should be performed.

The docking protocol used to predict the spike epitopes uses a C-alpha resolution to represent protein residues, and is data-driven, i.e. it assumes that binding happens in the first place, and then utilizes experimentally obtained structural restraints. So, concluding possible binding from such a docking protocol alone would be noisy. In our revised manuscript we have a new Figure 3B, which shows epitopes of 4 out of the 5 pan-VoC nanobodies, i.e. (S1-RBD-9, 22, 40) and S1-46 mapped to the RBD structures of XBB.1 (8IOU) and BQ.1.1 (8FXC), and we have updated Figure 4 with a supplemental showing the mutation maps.

(2) Described nanobodies are positioned as very potent neutralizers of SARS-CoV-2. However, they are much less potent in neutralization of ancestral strain as well as early VOCs compared to the mAbs that were approved for COVID-19 treatment. For example, IC50 for casirivimab and imdevimab are 37.4 pM and 42.1 pM, respectively. That is about 27-fold more than IC50 for the most potent nanobody reported in the article, S1-RDB-15.

This comparison is fraught for several reasons. 1. Experimental differences in pseudovirus assay systems usually result in significant differences in reported IC50s, as IC50 is not an absolute measure, or ultimately comparable to clinical IC50 values. For this reason, in our original publication (Mast et al., 2021) we tested other nanobodies in our experimental set-up as benchmarks (Mast et al., 2021). 2. A typical monoclonal has two binding sites with a large structural Fc linker that is combined ~10 times the size of a nanobody. In a therapeutic setting where monoclonal therapy is provided in g per kg of patient body weight, there is a 5-fold excess of Nb binding to antibody binding capacity. 3. We have previously shown that dimerizing our nanobodies (to produce two antigen binding sites) can dramatically increase potency over 100 fold (Mast et al., 2021).

In order to make this even clearer in the manuscript, we have added the following: “We note that IC50s are not directly comparable across different experimental set-ups because measured values are highly dependent on the experimental conditions. For this reason, we included other published nanobodies as benchmarks in our original publication and have subsequently maintained standard experimental conditions (Mast, Fridy et al. 2021)”.

(3) Figure 1A. If each dot represents an independent measurement of the same nanobody, IC50 variation seems too high. For some nanobodies it ranges for almost a log of magnitude, e.g S1-RDB-24, S1-RBD-46, S2-3. Why is that?

We have deliberately explored the full range of effects that could contribute to experimental variability in our pseudovirus assay, using different batches of nanobody and pseudovirus in each replicate to provide as impartial and comprehensive analysis as possible. While the activity of some nanobodies is remarkably stable from batch to batch, others show the variation noticed by the Reviewer, hence why we performed multiple replicates to define the average IC50 value for our nanobodies.

(4) The drop in IC50 for BA.1 neutralization is about one log for the majority of tested nanobodies. This should be outlined in the text. For example, for the most potent neutralizer, S1-RDB-15, the drop in IC50 for BA.1 is about 100-fold compared to IC50 for the Delta and Wuhan strains. It is important to note that out of 9 nanobodies for that drop in neutralizing capacity against BA.1 and Delta variants less than one log of magnitude 2 have epitopes in the S2 domain of SRS-CoV-2 spike. Resistance of mAbs targeting the S2 part of the spike has been extensively described in the literature as being due to the highly conserved structure of this region that facilitates membrane fusion. Presented data demonstrate that >80% of the nanobody repertoire is affected by mutations on spike protein. Additionally, it can be helpful for readers if the fold-change in IC50 between Wuhan, Delta, and BA.1 is presented in the text or added to Figure 1 or a table.

We agree with the Reviewer and to make this more explicit we have made the following change:“In comparison, groups I, I/II, I/IV, V, VII, VIII and the anti-S2 nanobodies contained the majority of omicron BA.1 neutralizers, though here the neutralization potency of many nanobodies was generally decreased tenfold compared to wild-type (emphasis added).”

(5) The authors should either present the results of the formal correlation analysis or avoid using misleading verbiage such as: "the decrease in neutralization potency largely correlates with the accumulation of omicron BA.1 specific mutations throughout the RBD" or "significant decrease in binding affinity correlated to decreases neutralization potency".

We thank the Reviewer for this constructive feedback. To address this question, we have performed a correlation analysis using Pearson and Spearman's methods to quantitatively assess the relationship between nanobody neutralization potency (IC50) and binding affinity (KD) across SARS-CoV-2 variants, including the wildtype, delta, and omicron BA.1 variants.Our results indicate a statistically significant correlation for the delta variant (Pearson's PCC: 0.71, p-value: 0.01; Spearman's rho: 0.63, p-value: 0.07), supporting our statement regarding the correlation between decreased neutralization potency and reduced binding affinity for this variant. However, for the wildtype and omicron BA.1 variants, the correlations were not statistically significant (wildtype Pearson's: 0.10, p-value: 0.70; omicron BA.1 Pearson's: 0.27, p-value: 0.31), which we acknowledge does not fully align with the verbiage used in the manuscript. Therefore, we have revised the manuscript to present the correlation analysis data accurately and ensure the discussion is reflective of the statistical evidence as follows:

“SPR binding assessments to the spike S1 domain or RBD of delta revealed a pattern: nanobodies maintaining binding affinity generally also neutralized the virus with a statistically significant correlation between binding affinity and neutralization efficacy (Pearson's Correlation Coefficient: 0.71, p-value: 0.01; Spearman's rho: 0.63, p-value: 0.07). However, this correlation was not statistically significant for omicron BA.1 (Pearson's Correlation Coefficient: 0.27, p-value: 0.31) (Fig. 3A, Table 1). Notably, while some nanobodies bound to the variants, they did not consistently neutralize them, suggesting additional factors influence neutralization beyond mere binding.”

(6) Figure 3 shows approximated curves for live virus neutralization assay with quite a broad 90% CI. It will be helpful to present, at least, in supplementary, primary data for live-virus neutralization that were used to perform non-linear regression.

We have added the reviewer’s suggestion.

(7) It is not clear what are the "variant-specific nanobody groups" exactly? A definition/description of the term is not provided. If the nanobody library was generated with the Wuhan strain, how did strain-specific nanobodies that bind/neutralize only Delta, BA.1 or BA.4/5 appear in the repertoire and were isolated? This statement also contradicts data in Table 4 where all nanobodies listed bind and neutralize Wuhan strain.

We agree with the reviewer. All nanobodies tested bind/neutralize the Wuhan strain as they were selected from our original repertoire of 116 nanobodies (Mast, et al., 2021). To clarify, variant-specific nanobodies are nanobodies that bind only one variant that arose from the original Wuhan strain. They were categorized into variant-specific groups based on whether they were able to bind each variant (other than Wuhan).

We have thus added to the manuscript, “we define variant-specific nanobodies as nanobodies that bind a single additional variant alongside the original Wuhan strain...”

(8) Describing the categorization of nanobody epitope groups presented in Figure 4, the authors state that binding to Wuhan, Delta, BA/1, and BA.4/5 predicts that these nanobodies will be "effective binders against current circulating strains of the virus including XBB and BQ lineages"? How exactly is this conclusion corollary to the data shown?

The epitopes of XBB and BQ.1 are not divergent enough within the regions we propose the nanobodies to bind, to suggest that nanobodies that bind in those regions will lose binding ability. We hypothesize that the region at which these nanobodies bind represents regions on spike that are vulnerable to our specified nanobodies in Fig. 4. We have generated a new Fig. 3B and added a supporting figure for Fig. 4 to address this.

(9) Figures 4C and 6 describe how the nanobodies will retain binding to currently circulating strains of XBB lineage. However, epitopes are mapped on the same Wuhan, Delta, BA.1, and BA.4/5 virus strains. The predicted binding of nanobodies to XBB lineage RBD is not actually shown in Figure 6. It is clear from the figure that the nanobody binding footprint (red area) decreases with antigenic distance in every spike projection from Wuhan through the BA.4/5 strain. It is unclear how this indicates that nanobodies will remain active against even more distant XBB, BQ, EU, and CH strains accumulating more mutations in spike protein.

We have added the following to the manuscript to clarify: “Strikingly, we have in our cohort 8 nanobodies able to bind delta, and the omicron lineages BA.1/BA.4/BA.5/XBB/BQ.1.1 (Fig. 5B). We further predict these 8 nanobodies will be effective binders against current circulating strains of the virus including omicron EG.5 and HV.1 as the epitope regions (or predicted epitopes) of these nanobodies do not vary significantly from omicron lineages XBB and BQ.1.1 (Fig. 5C and Supplement to Fig. 5).”

(10) Despite major advances in the development of nanobodies as therapeutic molecules there are only a few nanobody-based drugs that have so far been approved for clinical use and all of them are nanobody fusions to immunoglobulin Fc fragment. It is dictated by the small size of the nanobody itself, 15 kDa molecule, that leads to rapid kidney clearance within hours post-injection, and also by the necessity of having antibody effector functions allowing for example killing of malignant cells. It is hard to predict how each individual nanobody will tolerate multimerization and if it will still retain binding ability as its size dramatically increases. It should be noted that IC50 for BA.4/5 is in the submicromolar range for the 5 nanobodies retaining neutralization of this strain. From a therapeutic perspective, this is quite a high IC50 that dictates a high dosage to achieve a therapeutic effect. Furthermore, it can be expected that additional mutations in the SARS-CoV-2 spike will further affect binding affinity and therefore reduce the neutralization ability of these nanobodies resulting in even higher doses required to achieve therapeutic effect. Therefore, authors should discuss the limitations of the nanobody approach as a therapeutic intervention more granularly.

While Fc fusions are not strictly required for clinical use (for instance Caplacizumab is not an Fc fusion, being a multimer containing an albumin-binding nanobody), we agree that reformulation would indeed be required to optimize pharmacokinetics for eventual clinical use. Increased valency through multimerizeration is in fact one of several strategies, which also includes synergistic combinations, for significantly enhancing effective IC50. Preclinical nanobody engineering is not within the scope of this paper, but we acknowledge this challenge.

Minor points:(1) Table S1 is missing.

This is an .xlsx file uploaded as Supplementary File 3. Labeled now as “Figure 6–Source data 2. Neutralization data from synergy experiment”.

(2) Because Table 1 summarizes all neutralization and binding data, it will be helpful to refer to it while describing data presented in Figure 1.

This has been added to the revised manuscript.

(3) Live SARS-CoV-2 PRNT is not described in Materials and Methods.

This has been added to the revised manuscript.